# Development of an immuno-wall device for the rapid and sensitive detection of *EGFR* mutations in tumor tissues resected from lung cancer patients

Naoyuki Yogo[1,2,3], Tetsunari Hase[1,3]*, Toshihiro Kasama[3,4], Keine Nishiyama[5], Naoya Ozawa[1], Takahiro Hatta[1], Hirofumi Shibata[1], Mitsuo Sato[1,6], Kazuki Komeda[1], Nozomi Kawabe[6], Kohei Matsuoka[6], Toyofumi Fengshi Chen-Yoshikawa[7], Noritada Kaji[3,8], Manabu Tokeshi[3,9], Yoshinobu Baba[2,3,10], Yoshinori Hasegawa[1,11]

1 Department of Respiratory Medicine, Graduate School of Medicine, Nagoya University, Nagoya, Japan, 2 Department of Biomolecular Engineering, Graduate School of Engineering, Nagoya University, Nagoya, Japan, 3 Institute of Nano-Life-Systems, Institutes of Innovation for Future Society, Nagoya University, Nagoya, Japan, 4 Department of Bioengineering, School of Engineering, The University of Tokyo, Tokyo, Japan, 5 Graduate School of Chemical Sciences and Engineering, Hokkaido University, Sapporo, Japan, 6 Department of Pathophysiological Laboratory Sciences, Nagoya University Graduate School of Medicine, Nagoya, Japan, 7 Department of Thoracic Surgery, Nagoya University Graduate School of Medicine, Nagoya, Japan, 8 Department of Applied Chemistry, Graduate School of Engineering, Kyushu University, Fukuoka, Japan, 9 Division of Applied Chemistry, Faculty of Engineering, Hokkaido University, Sapporo, Japan, 10 Health Research Institute, National Institute of Advanced Industrial Science and Technology (AIST), Takamatsu, Japan, 11 National Hospital Organization, Nagoya Medical Center, Nagoya, Japan

* thase@med.nagoya-u.ac.jp

**Data Availability Statement:** All relevant data are within the paper and its Supporting information files.

## Abstract

Detecting molecular targets in specimens from patients with lung cancer is essential for targeted therapy. Recently, we developed a highly sensitive, rapid-detection device (an immuno-wall device) that utilizes photoreactive polyvinyl alcohol immobilized with antibodies against a target protein via a streptavidin–biotin interaction. To evaluate its performance, we assayed *epidermal growth factor receptor* (*EGFR*) mutations, such as E746_A750 deletion in exon 19 or L858R substitution in exon 21, both of which are common in non-small cell lung cancer and important predictors of the treatment efficacy of *EGFR* tyrosine kinase inhibitors. The results showed that in 20-min assays, the devices detected as few as 1% (E746_A750 deletion) and 0.1% (L858R substitution) of mutant cells. Subsequent evaluation of detection of the mutations in surgically resected lung cancer specimens from patients with or without *EGFR* mutations and previously diagnosed using commercially available, clinically approved genotyping assays revealed diagnostic sensitivities of the immuno-wall device for E746_A750 deletion and L858R substitution of 85.7% and 87.5%, respectively, with specificities of 100% for both mutations. These results suggest that the immuno-wall device represents a good candidate next-generation diagnostic tool, especially for screening of *EGFR* mutations.

**Funding:** This work was supported in part by the "Knowledge Hub Aichi" Priority Research Project from the Aichi Prefectural Government, the Nagoya University Hospital Funding for Clinical Research, and the Translational Research Network Program from the Japan Agency for Medical Research and Development (AMED).

**Competing interests:** T. Hase received personal fees from Chugai Pharmaceutical Co. Ltd., Ono Pharmaceutical Co. Ltd., Bristol-Myers Squibb Co., and Boehringer Ingelheim, and research funding from Boehringer Ingelheim, and Taiho Pharmaceutical Co. Ltd., outside the submitted work. Y. Hasegawa received grants from Boehringer Ingelheim, AstraZeneca, Eli Lilly Japan K.K., Ono Pharmaceutical Co. Ltd., Bristol-Myers Squibb Co., Taiho Pharmaceutical Co. Ltd., Novartis Pharma K. K., and Chugai Pharmaceutical Co. Ltd., and personal fee from Boehringer Ingelheim, MSD K.K., AstraZeneca, Pfizer Inc., and Chugai Pharmaceutical Co. Ltd, outside the submitted work. This does not alter our adherence to PLOS ONE policies on sharing data and materials.

## Introduction

Lung cancer is the leading cause of cancer-related mortality worldwide [1], and ~85% of lung cancers are classified as non-small cell lung cancer (NSCLC) [2]. *Epidermal growth factor receptor* (*EGFR*) is a member of the ErbB receptor tyrosine kinase family, which plays an important role in NSCLC cell proliferation, motility, and differentiation [3]. Somatic *EGFR* mutations are detected in 10% to 16% of NSCLC patients in the United States and Europe [4] and 30% to 50% of those in Asia [5], with ~90% presenting as deletions in exon 19, most commonly the E746_A750 deletion, and an L858R substitution in exon 21. Several studies report that these mutations are associated with the sensitivity of NSCLC patients to *EGFR* tyrosine kinase inhibitors (TKIs) [6,7]; therefore, clinical testing for *EGFR* mutations has become a standard care for patients with NSCLC. In addition to *EGFR* mutations, other driver mutations have been examined, with assessment of *anaplastic lymphoma kinase* (*ALK*), *c-ros oncogene 1* (*ROS1*), and *B-Raf proto-oncogene, serine/threonine kinase* (*BRAF*) currently part of routine molecular testing in Japan [8].

Direct sequencing of *EGFR* polymerase chain reaction (PCR) products is a common approach for *EGFR*-mutation testing [9]; however, its clinical usefulness is reduced by false-negative results due to the small proportion of cancer cells in collected samples available for DNA extraction. Other DNA-based analyses have been developed to detect *EGFR* mutations, including PCR-Invader [10] and peptide nucleic acid-locked nucleic acid (PNA-LNA) PCR clamp [11]. These methods show high sensitivity and can be used in patients with advanced NSCLC, even in those with low tumor-cell content [12]. However, routine testing using these methods is often limited by the associated high costs and technical complexity [9].

We previously developed various microfluidic immunoassay devices for rapid and highly sensitive molecular analyses. First, we proposed an immuno-pillar device [13] comprising antibody immobilized microbeads and a UV-curable polyethylene glycol-based resin cured using photolithography. Antibodies and antigens pass through pores in the cured resin to reach the microbeads, thereby allowing antibody–antigen reactions on the microbead surface. Utilizing sandwich-type immunoassays with accumulated three-dimensional (3D) fluorescence signals, immuno-pillar devices exhibit high biomarker detection sensitivity using human serum samples. However, floating substances, such as blood cells, cell debris, and fibrin, sometimes block the pores or persist near the microbeads and/or the cured resin after the immunoassay, resulting in false-negative or false-positive results. To solve this problem, we developed a new immunoassay device called an immuno-wall device from a non-porous photopolymer and that shows robust sensitivity, even in the presence of bodily fluids and lysed tumor tissue harboring large amounts of debris [14].

In the present study, we evaluated the ability of the immuno-wall device to specifically detect mutated EGFR proteins in surgically resected tissues from NSCLC patients and successfully performed rapid mutant EGFR detection in a small volume (1 μL) of lysed, debris-rich, surgically resected samples without the need for thorough pretreatments.

## Materials and methods

### Chemicals

We first prepared 1% (v/v) bovine serum albumin (BSA; Thermo Fisher Scientific, Waltham, MA, USA) in phosphate-buffered saline (PBS; Thermo Fisher Scientific) and PBS containing 1% (v/v) Tween-20 (PBS-T; Sigma-Aldrich, St. Louis, MO, USA). Washing buffer was prepared by mixing 1% BSA and 1% PBS-T (1:1; v/v). A photoreactive polyvinylalcohol (azido-unit pendant water-soluble photopolymer; AWP) for photo-immobilization was purchased

from Toyo Gosei Co., Ltd. (Tokyo, Japan). Recombinant streptavidin was purchased from ProSpec (Cat. No. pro-791NJ; East Brunswick, NJ, USA). Plastic immuno-wall device substrates were acquired from Sumitomo Bakelite Co., Ltd. (Tokyo, Japan). A rabbit anti-human EGFR (L858R) biotinylated antibody (clone 43B2, Cat. No. 5354), rabbit anti-human EGFR (E746_A750 deletion mutant) biotinylated antibody (clone D6B6, Cat. No. 5747), and rabbit anti-human EGFR biotinylated antibody (clone D38B1, Cat. No. 6627) were purchased from Cell Signaling Technology (Danvers, MA, USA). These antibodies were diluted to 50 μg/mL in PBS and used as the capture antibody in our sandwich immunoassay. We dissolved 50 μg of goat anti-human EGFR antibody (Cat. No. AF231; R&D Systems. Minneapolis, MN, USA) in 1% BSA (1 mL) and used it as the detection antibody in our sandwich immunoassay. These antibodies were also used for western blot analysis and immunocytochemistry. Importantly, anti-EGFR [wild-type; WT] antibodies can capture both WT and mutant EGFR. DyLight 650-conjugated anti-goat IgG antibody was purchased from Abcam (Cat. No. ab102343; Cambridge, UK) and diluted to 50 μg/mL with 1% BSA before use. Mouse monoclonal anti-actin antibody (Sigma-Aldrich) was used as a loading control, and an anti-rabbit or anti-mouse antibody (GE Healthcare, Little Chalfont, UK) was used as the secondary antibody for western blot analyses. Alexa Fluor 488 goat anti-rabbit IgG (Molecular Probes; Invitrogen, Carlsbad, CA, USA) was also used as a secondary antibody for immunocytochemistry, and 4',6-diamino-2-phenylindole (DAPI) (DOJINDO LABORATORIES, Kumamoto, Japan) was used for nuclei staining. Pipettes were used to inject the samples and reagents into the microchannels (Research Plus; Eppendorf AG, Hamburg, Germany). Solutions were removed from the microchannels with an aspirator (VACUSIP; INTEGRA Biosciences AG., Zizers, Switzerland).

## NSCLC cell lines and lysates

Human lung cancer cell lines H3255, and HCC827 were obtained from the Hamon Center Collection (University of Texas Southwestern Medical Center, Dallas, TX, USA) and H358, H1299, and PC9 were purchased from ATCC (Manassas, VA, USA). These cells were cultured in Roswell Park Memorial Institute 1640 medium supplemented with 10% FBS at 37°C in 5% $CO_2$. Cells were lysed with lysis buffer (Cell Signaling Technology) supplemented with 1 mM phenylmethylsulfonyl fluoride. All cell lines were checked for mycoplasma contamination using MycoAlert Mycoplasma Detection Kit purchased from Lonza (Cat. No. LT07-118; Switzerland) and short tandem repeats profiling analysis has been submitted for authentication.

## Resected tissues from NSCLC patients

Patients with pathologically confirmed NSCLC at Nagoya University Hospital between November 2010 and August 2015 were enrolled in this study. All participants provided written, informed consent, and the Ethics Review Committee of Nagoya University Graduate School of Medicine approved this study (No. 2014–0171). Surgically resected tumor tissues were preserved by snap-freezing in liquid nitrogen within 1 h of collection and stored at −80°C until use. Each tumor sample was divided into two pieces: one for the immuno-wall assay and the other for testing using a commercially available, clinically approved *EGFR* testing method (PNA-LNA PCR clamp or PCR-invader).

## Preparation of tumor lysate

The frozen tumor tissue was lysed in 200 μL radioimmunoprecipitation assay buffer (Wako, Osaka, Japan) with protease inhibitor using a sample-grinding kit (GE Healthcare). The lysate was centrifuged at 12,000$g$ for 10 min at 4°C, and supernatants were used for all assays.

## Western blot analysis

Western blot analysis was performed as described previously [15]. Primary antibodies used for this analysis included the anti-human EGFR (L858R) antibody (clone 43B2, Cat. No. 3197) (1:2,000), rabbit anti-human EGFR (E746_A750 deletion mutant) (clone D6B6, Cat. No. 2085) (1:5,000), anti-human EGFR antibody (clone D38B1, Cat. No. 4267) (1:4,000), and mouse monoclonal anti-actin antibody (Cat. No. A2228) (1:20,000). Horseradish peroxidase-conjugated donkey anti-rabbit (1:2,000) or sheep anti-mouse (1:2,000) antibody was used as the secondary antibody. Actin was used as a loading control.

## Immunocytochemistry

Lung cancer cells grown on plastic dishes were fixed and permeabilized for 30 min with 4% formaldehyde and 0.2% Triton X-100 in PBS. After blocking with 1% BSA for 1 h, cells were incubated with anti-human EGFR (L858R) antibody (clone 43B2, Cat. No. 3197) (1:1,000), rabbit anti-human EGFR (E746_A750 deletion mutant) (clone D6B6, Cat. No. 2085) (1:1,000), and anti-human EGFR antibody (clone D38B1, Cat. No. 4267) (1:1,000) in 1% BSA in PBS for 1 h at room temperature. Cells were then washed and incubated with Alexa Fluor 488 goat anti-rabbit secondary antibody (1:1,000) and DAPI (1:1,000) for 1 h at room temperature. Fluorescence images were obtained using an IX73 inverted microscope (Olympus, Tokyo, Japan) with a 10× objective lens.

## Immuno-wall devices

Immuno-wall device schematics are shown in Fig 1. The plastic substrates were made with cyclic olefin polymer, with 40 microchannels formed in the substrate. The wall-like structure with immobilized antibodies at the center of the microchannel was constructed using standard photolithography techniques. Devices were fabricated as follows. Streptavidin diluted in PBS (10 mg/mL) was mixed with the same volume of AWP in a low-adhesion tube (PROTEO-SAVE; Sumitomo Bakelite Co., Ltd., Tokyo, Japan) to prevent nonspecific streptavidin binding. The mixture was then introduced into the microchannels and irradiated with UV light (LA-410UV-5; Hayashi Watch-Works Co., Ltd., Tokyo, Japan) through a photomask. Streptavidin was also photo-immobilized onto the irradiated, cross-linked AWP, with uncured AWP containing free streptavidin removed using an aspirator. The AWP wall was constructed in the middle of the microchannels, which were washed and filled with washing buffer containing 0.5% BSA to prevent nonspecific analyte protein and antibody binding before use. The AWP wall extended from the microchannel floor to the roof. Therefore, the top and bottom surfaces of the AWP wall were not in direct contact with the loaded lysate sample or antibody solutions during the immunoassay.

To immobilize the capture antibody, 1 μL of biotinylated antibody in PBS was injected into the microchannel and incubated for 1 hour at room temperature. Most biotinylated antibodies were immobilized by streptavidin via a biotin–streptavidin interaction around the AWP wall, after which unbound antibodies were then removed. We named the antibody immobilized AWP wall structure the "immuno-wall."

## Immunoassay procedure

The immunoassay procedure was similar to that used for enzyme-linked immunosorbent assay (ELISA) using microtiter plates. Solutions were removed and injected using an aspirator or pipette, with the volume of the sample, antibody solution, and washing buffer at 1 μL/injection. First, we removed the washing buffer and injected the sample, and after incubation for 15

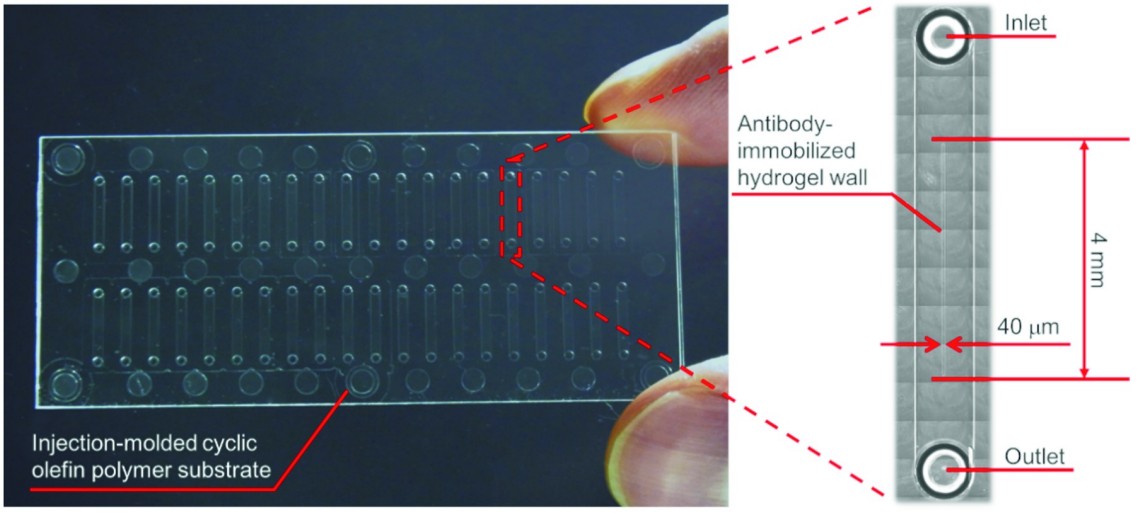

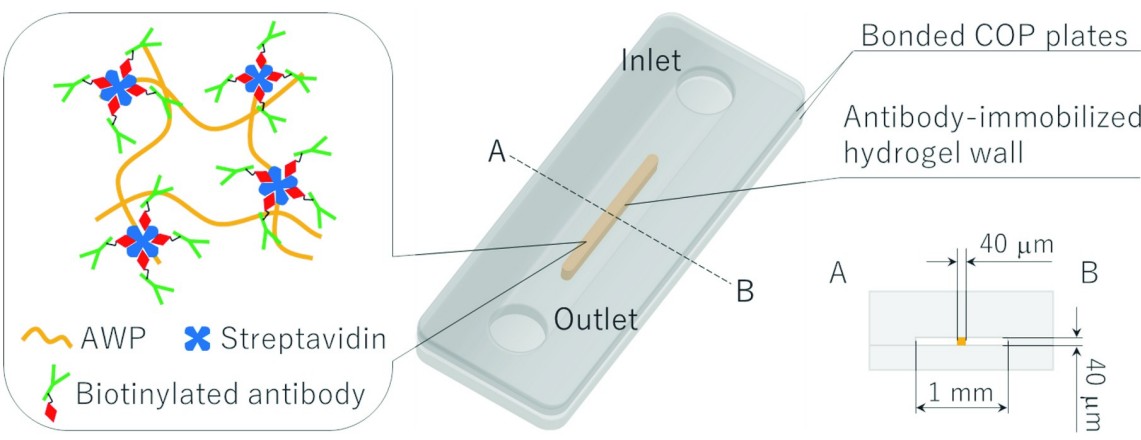

**Fig 1. Picture and schematic of the immuno-wall device.**

min, we removed the unreacted sample and washed by immersion (1 min) and sequential rinsing (15 times) with washing buffer. Cell debris remaining within the microchannel after sample incubation was removed by rinsing. The device was then immersed in detection antibody (30 s), washed by immersion (1 min) and 15 sequential rinses, and immersed in fluorescence-labeled anti-goat IgG antibody (30 s). The device was washed again and imaged using a fluorescence microscope (Nikon, Tokyo, Japan). The device was scanned by a fluorescence immunoassay reader (Hamamatsu Photonics K.K., Hamamatsu, Japan) according to manufacturer's instructions.

## Statistical analysis

Although no reference or 'gold standard' has been defined for *EGFR* mutation analysis, PCR-based methods are assumed to be the current gold standard in daily practice. Therefore, we calculated diagnostic sensitivity and specificity based on PCR-based results as described previously [16]. JMP pro 14 (SAS Institute Inc., Cary, NC, USA) was used for all statistical analyses in this study.

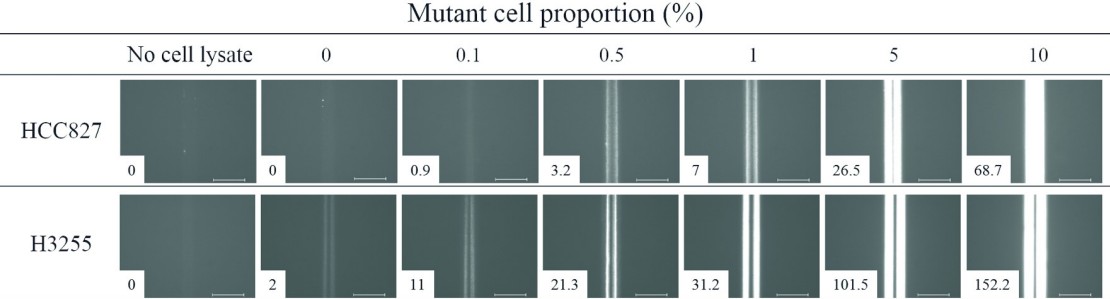

**Fig 2. Representative fluorescence images after immunoassays of lysates containing mixed populations of mutant *EGFR* cells.**
Fluorescence intensity measured by a fluorescence immunoassay reader is indicated. Scale bars = 100 μm.

## Results

### Immuno-wall assays

To evaluate the immuno-wall devices, we analyzed mixed NSCLC cell lysates harboring mutant and WT *EGFR*. *EGFR* mutant cells (HCC827 for E746_A750 deletion and H3255 for L858R substitution) and WT (H358) were mixed with 0%, 0.1%, 0.5%, 1%, 5%, and 10% mutant cells. Mixed cell lines were lysed in 200 μL lysis buffer with a 1.5 mg/mL total protein concentration. Respective anti-EGFR antibodies were used as capture antibodies, and representative fluorescence images of the immuno-wall devices are shown in Fig 2. Fluorescence intensity increased along with increasing mutant *EGFR* cell proportions. All images, except for those for 0%, 0.1%, and 10% HCC827 cells, appeared as two bright lines, implying that the antigen–antibody reaction mainly occurred on the device side, likely due to the large amount of immobilized capture antibodies. Weak fluorescence observed in 0% H3255 samples indicated cross-reactivity of anti-EGFR (L858R) antibody with WT EGFR, which was consistent with immunocytochemistry and western blot results (S1 Fig).

Fluorescence calibration curves of the devices generated using the mutant *EGFR* cells are shown in Fig 3. Each plot represents the average (mean ± standard error of the mean) fluorescence intensity of the immuno-wall. The limit of detection (LOD) was estimated at 1% and 0.1% for the E746_A750 and L858R mutant cell proportions, respectively, based on the threshold value calculated as three standard deviations above the signals observed in the analyses for lysate with 1.5 mg/mL protein concentration from the EGFR WT cell line (H358) that was applied for each immuno-wall. The background fluorescence for the H3255 cell line was slightly higher than that for the HCC827 cell line, possibly due to cross-reactivity of the anti-EGFR (L858R) antibody. To estimate potential cross-reactivity, we performed the immunoassay using lysates from only the *EGFR* WT cell line (H358) (S2A and S2B Fig), finding that the fluorescence intensities remained below the threshold. For L858R substitution antibody, western blot analysis revealed a weak band for L858R substitution in HCC827 cells, which also indicated cross-reactivity (S1 Fig). Therefore, immunoassay was performed using lysates from only HCC827 and observed that the fluorescence intensities remained below the threshold (S2C Fig). Additionally, we performed the immunoassay targeting only mutant *EGFR* cell lines (Fig 4), revealing LODs estimated at ~0.01 mg/mL for both mutations and resulting in signals at three standard deviations above the average for the *EGFR* WT cell line (H358). At high protein concentration (>1 mg/mL), the increased fluorescence signals overlapped.

During immuno-wall analysis, target proteins were identified after incubation for only 15 min. Because cell lysate contains several proteins that could potentially interfere with the

**3A**

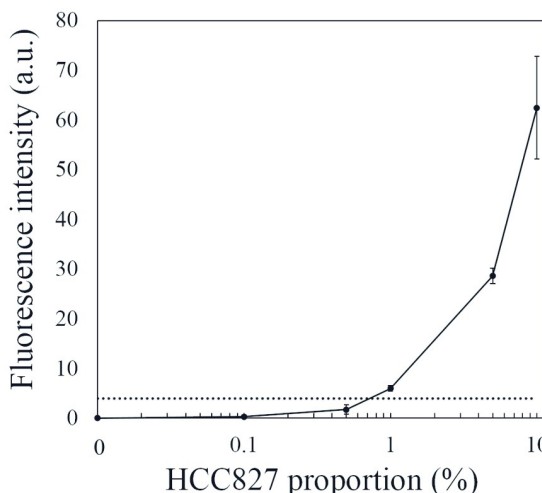

**3B**

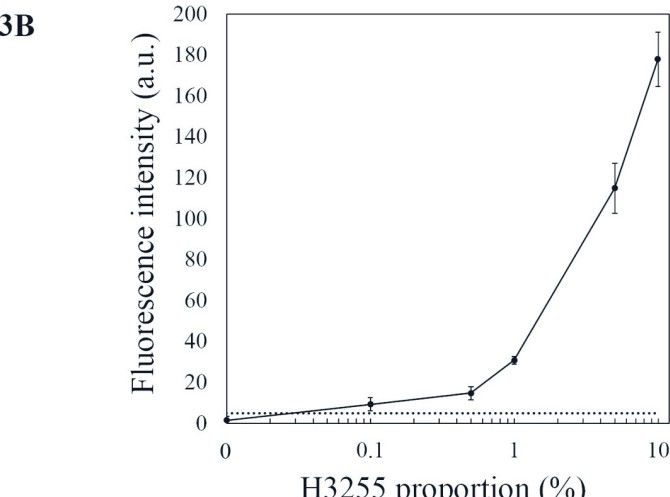

**Fig 3. Calibration curves derived from mutant *EGFR* cell lines.** Fluorescence intensities associated with (A) E746_A750 deletion and (B) L858R substitution versus background device fluorescence. The dashed lines indicate fluorescence values three standard deviations above the average for the *EGFR* WT cell line (H358) detected in each immuno-wall device.

antigen–antibody reaction, the capture-antibody density on the AWP wall might affect detection efficiency. We assumed that the biotinylated capture antibodies introduced into the microchannels would be immobilized at the AWP wall side surface when mixed with streptavidin. To evaluate immobilization efficiency, we compared direct and biotin–streptavidin immobilization to the AWP (S3 Fig). Naked or biotinylated goat IgG was injected into the device, and immobilization was detected by a fluorescence-labeled anti-goat IgG antibody. Representative fluorescence images showed increased fluorescence intensity in the biotin–streptavidin-binding device (S3A Fig), with fluorescence quantification showing a >10-fold higher fluorescence intensity in the device with biotin–streptavidin interactions (S3B Fig).

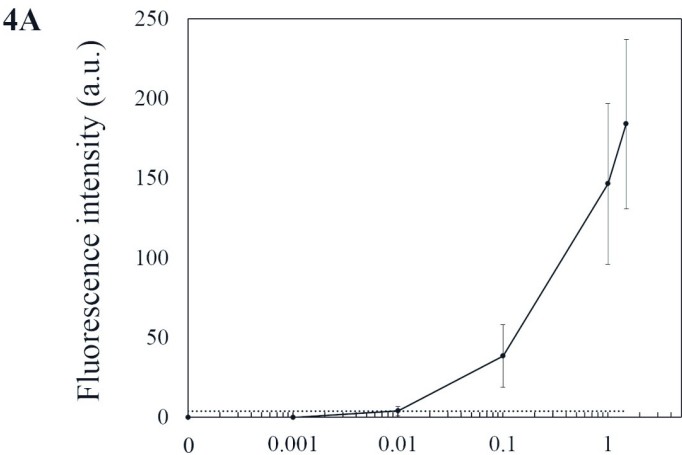

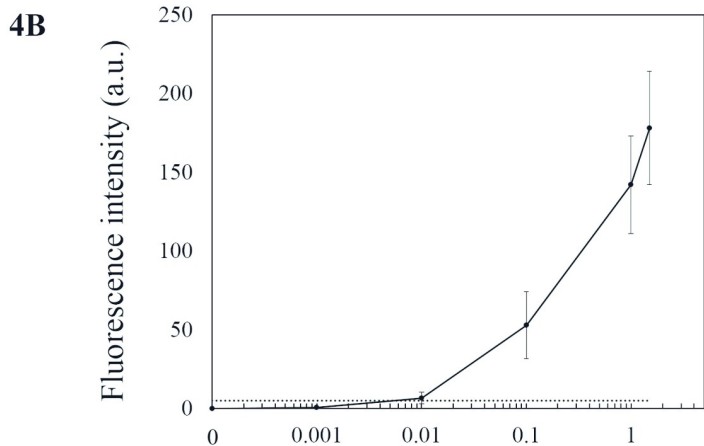

**Fig 4. Calibration curves derived from total protein concentration in mutant *EGFR* cell lines versus device fluorescence intensity.** The dashed lines indicate fluorescence values three standard deviations above the average for the *EGFR* WT cell line (H358) detected in each immuno-wall device.

## Clinical diagnostic application

The immuno-wall devices were capable of accurately analyzing small sample volumes (1 μL). The lysed samples from surgically resected tumors included a higher level of debris than that from cell-line samples (S4 Fig); however, pretreatments, such as thorough cell-debris removal and sample enrichment, were not required before the immunoassay, which allowed easy preparation and analysis of the clinical samples. The immuno-wall devices were then used to perform clinical diagnosis of surgically resected specimens from 22 NSCLC patients previously confirmed as harboring tumors with the E746_A750 deletion, L858R substitution, or *EGFR* WT according to PCR-based methods and the previously described threshold value (S1 Table). The results shown in Table 1 and Fig 5 indicated that immuno-wall analysis of the L858R

**Table 1. Genotype analysis of immuno-wall results.**

| Immuno-wall analysis | Genotype | | | |
|---|---|---|---|---|
| | E746_A750 deletion (*n* = 7) | Other exon 19 deletion (*n* = 15) | L858R substitution (*n* = 8) | WT (*n* = 7) |
| E746_A750 deletion, n (%) | 6 (85.7) | 0 | 0 | 0 |
| L858R substitution, n (%) | 0 | 0 | 7 (87.5) | 0 |
| WT, n (%) | 1 (14.3) | 15 (100) | 1 (12.5) | 7 (100) |

substitution demonstrated weak fluorescence using mixtures of lysates containing both *EGFR* E746_A750 deletion mutants and *EGFR* WT, which was consistent with in vitro findings (Fig 2). However, the weak fluorescence was lower than the threshold, resulting in a negative diagnosis. For lysates harboring both E746_A750 deletion and L858R substitution, the immuno-wall returned a >85% diagnostic sensitivity, with one case of each mutation misdiagnosed as negative due to low fluorescence intensity. Samples including *EGFR* WT were all diagnosed as such according to the immuno-wall device.

We then analyzed specimens from 15 NSCLC patients harboring an exon 19 deletion other than the E746_A750 deletion and confirmed by PCR-based methods (S1 Table and Table 1). All samples with the exon 19 deletion showed low fluorescence intensity when targeting the anti-E746_A750 deletion mutation and were diagnosed as WT based on detection of only the WT variant, resulting in 100% diagnostic specificity for both mutations.

## Discussion

Here, we describe fabrication of a new immunoassay device for highly sensitive and rapid detection of mutant *EGFR* variants in a small sample volume (1 μL) of lysed, surgically resected samples with containing high levels of cellular debris. For quantitative analysis, mixtures of lysed NSCLC cell lines (mutant and WT *EGFR*), resulting in an estimated LOD of 1% and

**Fig 5. Representative fluorescence images from immunoassay of clinical samples.** Samples containing (A) the E746_A750 deletion mutation (patient No. 11) and (B) L858R substitution mutation (patient No. 18). (C) A sample with no *EGFR* mutations (patient No. 9). Fluorescence intensity is indicated. Scale bars = 100 μm.

0.1% for the cell populations harboring the E746_A750 and L858R substitutions, respectively. Additionally, the LOD was estimated using a dilution series for each mutant *EGFR* cell line, revealing values as low as 0.01 mg/mL for both lines. Moreover, the diagnostic sensitivity and specificity of the immuno-wall device for the mutations were 85% and 100%, respectively. These results indicated that the device described here provided rapid and specific detection of *EGFR* mutations.

The immuno-wall device includes several features suitable for molecular assays, including employment of enhanced immobilization of capture antibodies using biotin–streptavidin binding (S3 Fig), thereby allowing a high probability of antigen capture. Compared with other immuno-assays, including western blot, the immuno-wall device employs a 3D reaction field at the side of the immuno-wall, where the fluorescence signals are integrated, which potentially increases the detection sensitivity of the device. Additionally, the wall-like structure enables easy removal of non-specifically bound molecules by injection of washing buffer. In general, specimens, including blood or lysed tissues, contain cell debris and fibrin (S4 Fig), which can disturb flow into and out of the microfluidic channel; therefore, removal of these items is important for microfluidic assays. However, the immuno-wall device is capable of accurately analyzing lysed tissue samples without the need for thorough pretreatment. Moreover, the limited area of the microchannel allowed rapid antigen–antibody interactions, resulting in shorter incubation times and rapid detection of target molecules. The results confirmed that the immunoassay procedure could be completed within 20 min, making this method suitable for clinical applications, including point-of-care diagnostics.

Recently, several studies examined the presence of *EGFR* mutations in lung cancer by immunohistochemistry (IHC) using the two mutation-specific antibodies employed in the present study and demonstrated sensitivity ranging from 24% to 100% and specificity ranging from 77% to 100% [16–23]. IHC is a well-established and cost-effective method routinely applied in lung cancer diagnosis; however, the results are sometimes affected by differences in assay procedures and scoring system [24]. In the present study, we established a fluorescence threshold based on the average fluorescence intensity of an *EGFR* WT cell line (H358) (Fig 3). Bellevicine et al. [25] demonstrated that IHC analysis using EGFR-mutant-specific antibody could detect 10% of mutated cells from a mixture containing cells harboring either WT or mutant *EGFR*. In the present study, our dilution series of each mutant *EGFR* cell line revealed an LOD estimated at between 0.1% and 1% (Fig 3), which meets the sensitivity requirement of CAP/IASLC/AMP guidelines promoting the use of more sensitive tests that can detect mutations in specimens with as few as 20% cancer cells [8]. Additionally, the immunoassay for lysates with high protein concentration from *EGFR* WT cell lines and specimens from NSCLC patients harboring *EGFR* WT showed fluorescence intensities below the threshold, suggesting a low probability of false positives, even at high protein concentrations. Moreover, the L858R substitution antibody might cross-react with WT *EGFR* or other *EGFR* mutations (S1 Fig). In previous studies where *EGFR* mutation was evaluated using IHC, false positive results were observed with the use of anti-EGFR (L858R) antibody [23,26]. To evaluate potential cross-reactivity, we performed the immunoassay using lysates from only HCC827, which were proven to be false positive by immunocytochemistry or western blotting (S1 Fig). We found that the fluorescence intensities remained below the threshold, indicating the robustness of the assay even in samples with confirmed cross-reactivity by the anti-EGFR (L858R) antibody. The importance of an absence of false positives in this situation is underscored by the ineffectiveness of *EGFR* TKIs in patients without *EGFR* mutations, which can result in early disease progression [27]. Although the sample size in the present study was small, the results demonstrated high diagnostic sensitivity and specificity for the device relative to those obtained by PCR-based assays.

The mutation-specific antibodies used in this study targeted only two types of representative *EGFR* mutations. Analysis of a sample harboring a different deletion mutation in exon 19 showed an extremely faint signal using the EGFR (E746_A750 deletion) antibody. Yu et al. [28] reported detection of the E746_T751 deletion mutation using an antibody targeting E746_A750 but not L747_ A750. Additionally, Kawahara et al. [23] reported that only two of seven minor deletion mutations in exon 19 were identified by the E746_A750 antibody, and Simonetti et al. [18] reported that 12 samples with minor deletions in exon 19 were undetectable by IHC using mutation-specific antibodies. These findings suggested that the clinical efficacy of some antibodies is reduced by the occurrence of certain similar mutations; however, because E746_A750 and L858R somatic mutations account for >70% of *EGFR* mutations [19], screening NSCLC patients using the device described in the present study could potentially increase diagnostic speed and accuracy and identification of candidates for *EGFR* TKI therapy. Further improvement of these mutation-specific antibodies is needed to detect other less frequent mutations in order to enhance the sensitivity of molecular diagnosis using our device as a tool for *EGFR* mutations screening.

Several factors may have been associated with the discordances between the PCR-based method and our procedure, even in the analysis of E746_A750 and L858R somatic mutations. Although the exact reasons were unknown, possible factors may have included variation in tumor cell content or tumor heterogeneity within samples, which were also observed in comparison studies for PCR-based testing [12,29–31].

This study has limitations. First, it was a retrospective, single-center study, in which mutation analysis was conducted using a limited number of surgical samples stored in the hospital. Therefore, selection bias might have affected the results. Second, direct comparison of IHC with the described method was not done due to the complexity of the staining procedure and the variability in scoring of staining intensity for IHC assays. Therefore, superiority over IHC was inconclusive. Additionally, direct comparison with ELISA was not conducted, because PCR is the current gold standard for *EGFR*-mutation analysis; therefore, we felt that this was the appropriate comparison. Moreover, ELISA generally requires several hours and higher sample volumes ($\geq$100 μL). Third, overlap in the fluorescence signal occurred at high protein concentrations (especially at $\geq$1 mg/mL protein) (Fig 4), indicating loss of quantifiable accuracy. In cases of highly concentrated samples harboring *EGFR* mutation(s), signal variability can potentially occur due to various forms of cell debris, including blood cells and fibrin. However, given the importance to detect lower proportions and lower concentrations of mutant molecules for this type of testing, overlaps of these signals at higher concentrations might be acceptable. Furthermore, many NSCLC patients are diagnosed with advanced disease and not surgically treated. Because small biopsy or cytological samples are only collected for diagnosis and mutation testing for these patients, such specimens are not always stored for other purposes. Therefore, these samples were not examined in this study. Taken together, our findings need to be confirmed in a larger prospective multicentric study.

## Conclusions

Here, we analyzed surgical samples from 37 NSCLC patients using a new immuno-wall device, revealing a diagnostic sensitivity and specificity of >85% and 100%, respectively. Moreover, samples could be prepared within 10 minutes, and the immunoassay procedure could be completed within 20 min, for a 30-min total assay time. These results suggest the immuno-wall device as a good candidate for next-generation diagnostics. Future studies will investigate the applicability of the device for small specimens or other molecular targets, such as *ALK*, *ROS1*, and *BRAF*.

## Supporting information

**S1 Fig. Expression of *EGFR* mutations (E746_A750 deletion and L858R substitution) in NSCLC cell lines.** (A) IHC analysis of each NSCLC cell line for the indicated antibody (green) and nuclei (blue). Images were obtained using a fluorescence microscope with a 10× objective lens. Scale bars = 10 μm. (B) Western blot analyses of mutant EGFR protein in each NSCLC cell line. Actin was used as a loading control.
(TIF)

**S2 Fig. Calibration curves of total protein concentration of lysates of *EGFR* WT cells (H358) and *EGFR* mutation cells (HCC827) versus the fluorescence intensity for each EGFR mutant protein.** E746_A750 deletion for *EGFR* WT cell line (H358) (A), L858R substitution for *EGFR* WT cell line (H358) (B), and *EGFR* mutation cells (HCC827) (C). Dashed lines indicate three standard deviations above the average fluorescence determined in the *EGFR* WT cell line (H358) by each immuno-wall device.
(TIF)

**S3 Fig. Comparison of immobilization to the AWP.** Goat IgG in PBS (100 μg/mL) and streptavidin in PBS (10 mg/mL) were mixed with an equal volume of AWP, respectively, introduced into the microchannel, and irradiated with UV light through a photomask. Biotinylated goat IgG (50 μg/mL) was injected into the microchannel, mixed with streptavidin, and the device was incubated for 60 min at room temperature, followed by the washing procedure. To compare immobilization to the AWP, fluorescence-labeled anti-goat IgG antibody was introduced into the microchannels of each device and incubated for 5 min at room temperature. After washing, (A) fluorescence images of the immuno-wall were obtained using a fluorescence microscope with a 20× objective lens (exposure time: 0.25 s) and (B) scanned with a fluorescence reader. Scale bars = 100 μm.
(TIF)

**S4 Fig. A representative microscopy image of cell lysates from NSCLC cell lines and tumor samples.** Scale bars = 100 μm.
(TIF)

**S1 Table. Details of patients examined and immuno-wall analyses.**
(DOCX)

## Acknowledgments

We thank Chiharu Okajima, Miwa Ito, and Yuko Mizuno for their technical support. We would like to thank Editage (www.editage.com) for English-language editing.

## Author Contributions

**Conceptualization:** Tetsunari Hase, Toshihiro Kasama.

**Data curation:** Naoyuki Yogo, Tetsunari Hase, Toshihiro Kasama, Keine Nishiyama, Naoya Ozawa, Takahiro Hatta, Hirofumi Shibata, Kazuki Komeda, Nozomi Kawabe, Kohei Matsuoka, Manabu Tokeshi.

**Formal analysis:** Naoyuki Yogo, Tetsunari Hase, Hirofumi Shibata, Kazuki Komeda, Nozomi Kawabe, Kohei Matsuoka.

**Funding acquisition:** Tetsunari Hase.

**Investigation:** Naoyuki Yogo, Tetsunari Hase, Toshihiro Kasama, Naoya Ozawa, Takahiro Hatta.

**Methodology:** Tetsunari Hase, Toshihiro Kasama, Keine Nishiyama, Mitsuo Sato, Toyofumi Fengshi Chen-Yoshikawa, Noritada Kaji, Manabu Tokeshi, Yoshinobu Baba.

**Project administration:** Tetsunari Hase, Toshihiro Kasama, Keine Nishiyama.

**Resources:** Tetsunari Hase.

**Supervision:** Tetsunari Hase, Toshihiro Kasama, Mitsuo Sato, Toyofumi Fengshi Chen-Yoshikawa, Noritada Kaji, Manabu Tokeshi, Yoshinobu Baba, Yoshinori Hasegawa.

**Validation:** Tetsunari Hase.

**Writing – original draft:** Naoyuki Yogo, Tetsunari Hase, Toshihiro Kasama.

**Writing – review & editing:** Tetsunari Hase, Toshihiro Kasama, Keine Nishiyama, Naoya Ozawa, Takahiro Hatta, Mitsuo Sato, Toyofumi Fengshi Chen-Yoshikawa, Manabu Tokeshi, Yoshinobu Baba, Yoshinori Hasegawa.

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
