## [Decision Letter · Decision Letter 0]

25 Aug 2020

PONE-D-20-01970

Development of an immuno-wall device for the rapid and sensitive detection of EGFR mutations in tumor tissues resected from lung cancer patients

PLOS ONE

Dear Dr. Hase,

Thank you for submitting your manuscript to PLOS ONE. After careful consideration, we feel that it has merit but does not fully meet PLOS ONE’s publication criteria as it currently stands. Therefore, we invite you to submit a revised version of the manuscript that addresses the points raised during the review process.

We look forward to receiving your revised manuscript.

Kind regards,

Srikumar Chellappan

Academic Editor

PLOS ONE

Additional Editor Comments:

The manuscript has been reviewed by two experts in the field; it has been difficult to recruit reviewers for this manuscript, for various reasons. My apologies. The comments are generally favorable, and please address them in a revised manuscript. Please ignore comments regarding patents etc. I look forward to seeing the revised submission.

2. Please provide additional information about each of the cell lines used in this work.

Specifically, please provide the specific source for each cell line used and and any quality control testing procedures (authentication, characterisation, and mycoplasma testing).

For more information, please see http://journals.plos.org/plosone/s/submission-guidelines#loc-cell-lines

3. Thank you for including your ethics statement: 

'All participants provided written, informed consent, and the hospital institutional review board approved this study (No. 2014-0171)'.    

a.Please amend your current ethics statement to include the full name of the ethics committee/institutional review board(s) that approved your specific study.  

b.Once you have amended this/these statement(s) in the Methods section of the manuscript, please add the same text to the “Ethics Statement” field of the submission form (via “Edit Submission”).

For additional information about PLOS ONE ethical requirements for human subjects research, please refer to " ext-link-type="uri" xlink:type="simple">http://journals.plos.org/plosone/s/submission-guidelines#loc-human-subjects-research."

4. At this time, we ask that you provide the product numbers and any lot numbers of the EGFR (L858R), EGFR (E746), EGFR and actin primary antibodies used in the Western blot analysis and immunocytochemistry analysis in this study.

5. To comply with PLOS ONE submission guidelines, in your Methods section, please provide additional information regarding your statistical analyses. For more information on PLOS ONE's expectations for statistical reporting, please see https://journals.plos.org/plosone/s/submission-guidelines.#loc-statistical-reporting

6. Thank you for stating the following in the Competing Interests section:

'T. Hase received personal fees from Chugai Pharmaceutical Co. Ltd., Ono Pharmaceutical Co. Ltd., Bristol-Myers Squibb Co., and Boehringer Ingelheim, and research funding from Boehringer Ingelheim, and Taiho Pharmaceutical Co. Ltd., outside the submitted work. Y. Hasegawa received grants from Boehringer Ingelheim, AstraZeneca, Eli Lilly Japan K.K., Ono Pharmaceutical Co. Ltd., Bristol-Myers Squibb Co., Taiho Pharmaceutical Co. Ltd., Novartis Pharma K. K., and Chugai Pharmaceutical Co. Ltd., and personal fee from Boehringer Ingelheim, MSD K.K., AstraZeneca, Pfizer Inc., and Chugai Pharmaceutical Co. Ltd, outside the submitted work.'

a. Please confirm that this does not alter your adherence to all PLOS ONE policies on sharing data and materials, by including the following statement: "This does not alter our adherence to  PLOS ONE policies on sharing data and materials.” (as detailed online in our guide for authors http://journals.plos.org/plosone/s/competing-interests).  If there are restrictions on sharing of data and/or materials, please state these.

Please note that we cannot proceed with consideration of your article until this information has been declared.

Reviewers' comments:

Reviewer's Responses to Questions

**Comments to the Author**

1. Is the manuscript technically sound, and do the data support the conclusions?

Reviewer #1: Partly

Reviewer #2: Partly

2. Has the statistical analysis been performed appropriately and rigorously? 

Reviewer #1: I Don't Know

Reviewer #2: No

3. Have the authors made all data underlying the findings in their manuscript fully available?

Reviewer #1: Yes

Reviewer #2: Yes

4. Is the manuscript presented in an intelligible fashion and written in standard English?

Reviewer #1: No

Reviewer #2: Yes

5. Review Comments to the Author

Reviewer #1: PLOS ONE

MS: PONE-D-20-01970

Development of immune-wall device for the rapid and sensitive detection of EGFR mutations in tumor tissues resected from lung cancer patients.

The research article by Yogo et al., 2020, developed a sandwich immunoassay based immuno-wall diagnostic tool to detect two prevalent EGFR mutants such as E746_A750 deletion in Exon 19 and L858R substation in exon 21. Both of these mutants are commonly observed in EGFR mutant NSCLCs and are good predictors of treatment efficacy upon EGFR TK inhibitors targeted therapy. Using both wildtype and mutant EGFR NSCLC cell lines and 37 lung cancer specimens, authors demonstrate that immuno-wall device detected the EGFR mutants E746_A750 and L858R mutants with sensitivities of 85.7% and 87.5% respectively.

Existing literature clearly reveal that immuno-wall device based detection of mutations in cancer has been published previously. An immuno-wall device for detection of IDH1-R132H mutation in low grade gliomas was described earlier (Yamamichi et al., 2016). Previously published article by Yogo et al., 2018 (same group as in the current paper) has already reported about the immuno-wall device for detecting Anaplastic lymphoma kinase 1 (ALK1) and C-ROS Oncogene 1(ROS1) fusion in lung cancer specimens. Thus, the development of the immuno-wall tool to detect EGFR mutants is relatively not novel and is merely an extension of the earlier research articles with replacement EGFR gene and its variants.

However, the scope of the study to identify the subset of NSCLC patients with EGFR mutations using immune-wall device have the potential to improve the therapeutic outcome and prognosis, as these mutations are associated with sensitivity to EGFR tyrosine kinase inhibitors.

Here are few concerns authors need to clarify.

Q1. In the figure S1B, lane 3, L858R substitution antibody picked up a less intense band in HCC827 (E746_A750del) cells but not in PC9 cells (E746_A750del). Whether HCC827 cells also carry L858R substitution. PC9 cells did not show any detectable band with L858R substitution antibody and thus, the PC9 cells should have been considered instead of HCC827.

Q2. The study tested very limited number of retrospective lung cancer specimens (N=37) for which DNA sequence for EGFR mutants were available. The test need to be performed on a large number of lung cancer specimens from multicentric studies and the revised sensitivity and specificity for each mutant should be considered as the diagnostic potential of this device. In the supplementary table 1, patient number # 14, (Female, age 51), the tumor lysate has the protein concentration of 6.32 µg/µl and carry E746_A750del as reported by DNA sequence. However, Immuno-wall device failed to detect the mutation in the specimen from the same patient (false negative). Did authors verify E746_A750del mutation in this patient specimen by western blot analysis.

Q3. In results section, please clearly explain and rewrite about estimation of limit of detection (LOD)

Reviewer #2: Reviewer’s Comments:

The present study by Yogo et al titled “Development of an immune-wall ………. Lung cancer patients” demonstrates an innovation comprises of a unique fluorescence-based immune-wall device, capably detecting a couple of frequently occurring EGFR mutations from both the cell line lysates and patient-derived samples in a very rapid, efficient and above all, cost-effective way; even from a very tiny amount of surgically resected and/or crude protein extracts. This is principally a biotin-streptavidin conjugated method to amplify fluorescence signals initiated by a basic epitope-based antigen-antibody interaction; which authors claim to be much better than the gold standard technic in the field, the conventional PCR-based detection procedure against these mutations. There are few pertinent questions this reviewer still possesses, which need better justification;

1. Though it is a rapid fluorescence-based technic, then how did authors claim this to be very unique; and above all, how it can be remarkably sensitive over conventional ICC/IHC methods?

2. During the assay, it looks like that the rinsing of cellular debris after the antigen-antibody reaction might be a critical step, failure of which can lead to false detection leading towards therapeutic mishaps. How do authors address this uncertainty?

3. The major problem of this entire study is the sample size; an overall 37 samples are not sufficient to prove their claim. This reviewer would have been satisfied if the patient sample size could have been at least 10-fold more with a multi-institutional level of verification via multiple hands.

4. Why H358 (EGFR WT) cells show a definite false positive signal utilizing the EGFR L858R antibody under Fig.S1A, in the lower panel? In such a scenario, how did authors claim their immune-wall detection methodology to be free from any significant errors; that’s why limiting sample size is always deceptive!

5. Why can’t authors patent their innovation before publishing it?

6. PLOS authors have the option to publish the peer review history of their article (what does this mean?). If published, this will include your full peer review and any attached files.

Reviewer #1: No

Reviewer #2: **Yes: **B. Saha, PhD

---

## [Author Response · Author response to Decision Letter 0]

9 Oct 2020

Response to Journal requirements and Reviewers’ Comments

We would like to thank the reviewers for their constructive comments. Our responses to all of the comments are provided below.

Journal requirements

Response: We have confirmed that this manuscript meets the style requirements of PLOS ONE.

2. Please provide additional information about each of the cell lines used in this work.

Specifically, please provide the specific source for each cell line used and any quality control testing procedures (authentication, characterization, and mycoplasma testing).

Response: We have revised the relevant sentences in the Methods section as suggested. (Page 7 line 115 to page 8 line 122)

Before: “Human lung cancer cell lines H358, H3255, H1299, PC9, and HCC827 were obtained from the Hamon Center Collection (University of Texas Southwestern Medical Center, Dallas, TX, USA) or purchased from ATCC (Manassas, VA, USA). These cells were cultured in Roswell Park Memorial Institute 1640 medium supplemented with 10% FBS at 37 °C in 5% CO2. Cells were lysed with lysis buffer (Cell Signaling Technology) supplemented with 1 mM phenylmethylsulfonyl fluoride.”

After: “Human lung cancer cell lines H3255, and HCC827 were obtained from the Hamon Center Collection (University of Texas Southwestern Medical Center, Dallas, TX, USA) and H358, H1299, and PC9 were purchased from ATCC (Manassas, VA, USA). These cells were cultured in Roswell Park Memorial Institute 1640 medium supplemented with 10% FBS at 37 °C in 5% CO2. Cells were lysed with lysis buffer (Cell Signaling Technology) supplemented with 1 mM phenylmethylsulfonyl fluoride. All cell lines were checked for mycoplasma contamination using MycoAlert Mycoplasma Detection Kit purchased from Lonza (Cat. No. LT07-118; Switzerland) and short tandem repeats profiling analysis has been submitted for authentication.”

3. Thank you for including your ethics statement: 

'All participants provided written, informed consent, and the hospital institutional review board approved this study (No. 2014-0171)'. 

Response: We have revised the relevant sentences as follows:

Page 8 line 126 to line 128 (revised portion is in red font).

“All participants provided written, informed consent, and the Ethics Review Committee of Nagoya University Graduate School of Medicine approved this study (No. 2014-0171).”

Response: As suggested, we have added the relevant sentence to the “Ethics Statement” field of the submission form.

4. At this time, we ask that you provide the product numbers and any lot numbers of the EGFR (L858R), EGFR (E746), EGFR and actin primary antibodies used in the Western blot analysis and immunocytochemistry analysis in this study.

Response: We have revised the relevant sentences as follows:

Page 9 line 139 to line 145 (revised portion is in red font).

“Primary antibodies used for this analysis included the anti-human EGFR (L858R) antibody (clone 43B2, Cat. No. 3197) (1:2,000), rabbit anti-human EGFR (E746_A750 deletion mutant) (clone D6B6, Cat. No. 2085) (1:5,000), anti-human EGFR antibody (clone D38B1, Cat. No. 4267) (1:4,000), and mouse monoclonal anti-actin antibody (Cat. No. A2228) (1:20,000). Horseradish peroxidase-conjugated donkey anti-rabbit (1:2,000) or sheep anti-mouse (1:2,000) antibody was used as the secondary antibody. Actin was used as a loading control.”

Page 9 line 149 to line 153 (revised portion is in red font).

“After blocking with 1% BSA for 1 h, cells were incubated with anti-human EGFR (L858R) antibody (clone 43B2, Cat. No. 3197) (1:1,000), rabbit anti-human EGFR (E746_A750 deletion mutant) (clone D6B6, Cat. No. 2085) (1:1,000), and anti-human EGFR antibody (clone D38B1, Cat. No. 4267) (1:1,000) in 1% BSA in PBS for 1 h at room temperature.”

5. To comply with PLOS ONE submission guidelines, in your Methods section, please provide additional information regarding your statistical analyses. 

Response: We added statistics information under the Methods section.

Page 12 lines 193 to line 197 (revised portion is in red font).

“Statistical analysis

Although no reference or ‘gold standard’ has been defined for EGFR mutation analysis, PCR-based methods are assumed to be the current gold standard in daily practice. Therefore, we calculated diagnostic sensitivity and specificity based on PCR-based results as described previously [16]. JMP pro 14 (SAS Institute Inc., Cary, NC, USA) was used for all statistical analyses in this study.”

6. Thank you for stating the following in the Competing Interests section:

'T. Hase received personal fees from Chugai Pharmaceutical Co. Ltd., Ono Pharmaceutical Co. Ltd., Bristol-Myers Squibb Co., and Boehringer Ingelheim, and research funding from Boehringer Ingelheim, and Taiho Pharmaceutical Co. Ltd., outside the submitted work. Y. Hasegawa received grants from Boehringer Ingelheim, AstraZeneca, Eli Lilly Japan K.K., Ono Pharmaceutical Co. Ltd., Bristol-Myers Squibb Co., Taiho Pharmaceutical Co. Ltd., Novartis Pharma K. K., and Chugai Pharmaceutical Co. Ltd., and personal fee from Boehringer Ingelheim, MSD K.K., AstraZeneca, Pfizer Inc., and Chugai Pharmaceutical Co. Ltd, outside the submitted work.'

a. Please confirm that this does not alter your adherence to all PLOS ONE policies on sharing data and materials, by including the following statement: "This does not alter our adherence to PLOS ONE policies on sharing data and materials.” (as detailed online in our guide for authors http://journals.plos.org/plosone/s/competing-interests). If there are restrictions on sharing of data and/or materials, please state these. 

Please note that we cannot proceed with consideration of your article until this information has been declared.

Response: We have confirmed that the competing interests do not alter our adherence to all PLOS ONE policies on sharing data and materials. Additionally, we have included the updated Competing Interests statement to our cover letter as suggested.

Review Comments to the Author

Reviewer #1

Q1. In the figure S1B, lane 3, L858R substitution antibody picked up a less intense band in HCC827 (E746_A750del) cells but not in PC9 cells (E746_A750del). Whether HCC827 cells also carry L858R substitution. PC9 cells did not show any detectable band with L858R substitution antibody and thus, the PC9 cells should have been considered instead of HCC827.

Response: Thank you for your valuable comment. We have confirmed that HCC827 harbors E746_A750del, not L858R, by peptide nucleic acid-locked nucleic acid PCR clamp as shown below, indicating cross-reactivity of anti-EGFR (L858R) antibody. The cross-reactivity is an important factor in the immunoassay including the immuno-wall device. In previous studies evaluating EGFR mutation in IHC, false positive results were also observed with the use of anti-EGFR (L858R) antibody[1,2]. As suggested, PC9 should have been considered instead of HCC827; however, we believe that it might be important to show no false positive results even in samples with confirmed cross-reactivity in the immuno-wall assay. In S2 figure, we performed immunoassay using lysates from only the EGFR WT cell line (H358) to evaluate potential cross-reactivity. In addition, we also performed the immunoassay using lysates from only the HCC827 and found that the fluorescence intensities remained below the threshold, indicating the robustness of the assay even in samples with confirmed cross-reactivity by anti-EGFR (L858R) antibody.

Together with the addition of a new supplementary figure S2C, we have revised and included the following sentences as shown below. 

Page 13 line 225 to page 13 line 229 (revised portion is in red font).

“For L858R substitution antibody, western blot analysis revealed a weak band for L858R substitution in HCC827 cells, which also indicated cross-reactivity (S1 Fig). Therefore, immunoassay was performed using lysates from only HCC827 and observed that the fluorescence intensities remained below the threshold (S2C Fig).”

Page 19 line 322 to line 329 (revised portion is in red font).

“Moreover, the L858R substitution antibody might cross-react with WT EGFR or other EGFR mutations (S1 Fig). In previous studies where EGFR mutation was evaluated using IHC, false positive results were observed with the use of anti-EGFR (L858R) antibody [23,26]. To evaluate potential cross-reactivity, we performed the immunoassay using lysates from only HCC827, which were proven to be false positive by immunocytochemistry or western blotting (S1 Fig). We found that the fluorescence intensities remained below the threshold, indicating the robustness of the assay even in samples with confirmed cross-reactivity by the anti-EGFR (L858R) antibody.”

Page 30 line 504 to line 509 (revised portion is in red font).

“S2 Fig. Calibration curves of total protein concentration of lysates of EGFR WT cells (H358) and EGFR mutation cells (HCC827) versus the fluorescence intensity for each EGFR mutant protein. E746_A750 deletion for EGFR WT cell line (H358) (A), L858R substitution for EGFR WT cell line (H358) (B), and EGFR mutation cells (HCC827) (C). Dashed lines indicate three standard deviations above the average fluorescence determined in the EGFR WT cell line (H358) by each immuno-wall device.”

Q2. The study tested very limited number of retrospective lung cancer specimens (N=37) for which DNA sequence for EGFR mutants were available. The test need to be performed on a large number of lung cancer specimens from multicentric studies and the revised sensitivity and specificity for each mutant should be considered as the diagnostic potential of this device. In the supplementary table 1, patient number # 14, (Female, age 51), the tumor lysate has the protein concentration of 6.32 µg/µl and carry E746_A750del as reported by DNA sequence. However, Immuno-wall device failed to detect the mutation in the specimen from the same patient (false negative). Did authors verify E746_A750del mutation in this patient specimen by western blot analysis.

Response: Thank you for your valuable comment. As suggested, this test must be evaluated with a larger number of lung cancer specimens in a multicentric manner; however, lung cancer specimens were not collected routinely, especially in community hospitals. Therefore, it was challenging to increase sample size in this retrospective study. Thus, a larger prospective multicenter study could be considered to confirm the result of the current study. 

Regarding patient #14, although a western blot analysis was performed, we failed to show the presence of E746_A750del mutation protein as shown below. The exact reason for the discordance between the DNA sequences and our method was unknown; however, possible factors may have included variation in tumor cell content or tumor heterogeneity within samples, which were also observed in comparison studies for PCR-based testing[3–6]. 

We have revised the relevant sentences as shown below. 

Page 22, line 369 to line 370 (revised portion is in red font).

“Taken together, our findings need to be confirmed in a larger prospective multicentric study.”

Page 20 line 348 to page 21 line 352 (revised portion is in red font).

“Several factors may have been associated with the discordances between the PCR-based method and our procedure, even in the analysis of E746_A750 and L858R somatic mutations. Although the exact reasons were unknown, possible factors may have included variation in tumor cell content or tumor heterogeneity within samples, which were also observed in comparison studies for PCR-based testing [12,29–31].”

Q3. In results section, please clearly explain and rewrite about estimation of limit of detection (LOD)

Response: Thank you for your valuable comment. The limit of detection (LOD) was estimated at 1% and 0.1% for the E746_A760 and L858R mutant cell proportions, respectively, based on the threshold value calculated as three standard deviations above the signals observed in the analyses for lysate with 1.5 mg/mL protein concentration from the EGFR WT cell line (H358) that was applied for each immuno-wall.

We have revised the relevant sentences as shown below. 

Page 13 line 218 to line 221 (the newly added part is in red font).

“The limit of detection (LOD) was estimated at 1% and 0.1% for the E746_A760 and L858R mutant cell proportions, respectively, based on the threshold value calculated as three standard deviations above the signals observed in the analyses for lysate with 1.5 mg/mL protein concentration from the EGFR WT cell line (H358) that was applied for each immuno-wall. ”

Reviewer #2

1. Though it is a rapid fluorescence-based technic, then how did authors claim this to be very unique; and above all, how it can be remarkably sensitive over conventional ICC/IHC methods?

Response: Thank you for your valuable comment. ICC/IHC method is a well-established and cost-effective method routinely applied in lung cancer diagnosis; however, the results are sometimes affected by differences in assay procedures and scoring system[7]. Indeed, several studies have examined the presence of EGFR mutations in lung cancer by IHC using the two mutation-specific antibodies employed in the present study and demonstrated sensitivity ranging from 24% to 100% and specificity ranging from 77% to 100%[1,8–14]. In addition, the cross-reactivity of the antibodies employed had potential risk for false-positive results in ICC/IHC methods. In our study, although there was concern regarding cross-reactivity of the anti-EGFR (L858R) antibody, no false positive results were obtained in the L858R assay owing to the pre-specified threshold, indicating not only rapidity, but also the usefulness of our method. 

2. During the assay, it looks like that the rinsing of cellular debris after the antigen-antibody reaction might be a critical step, failure of which can lead to false detection leading towards therapeutic mishaps. How do authors address this uncertainty?

Response: Thank you for your valuable comment. As mentioned, the rinsing of the debris following the antigen-antibody reaction was an important step in this assay. Compared with an immuno-pillar device, which was previously proposed in microfluidic immunoassays[15], an immuno-wall device utilizes a wall-like structure immobilized with antibodies at the center of the microchannel, which enabled easy removal of non-specifically bound molecules by the injection of washing buffer. These structural features might decrease the uncertainty, resulting in the diagnostic sensitivity and specificity of the immuno-wall device. 

3. The major problem of this entire study is the sample size; an overall 37 samples are not sufficient to prove their claim. This reviewer would have been satisfied if the patient sample size could have been at least 10-fold more with a multi-institutional level of verification via multiple hands.

Response: Thank you for your valuable comment. We agree with your comment. As described in our response to the comment of reviewer 1, lung cancer specimens were not collected routinely, especially in community hospitals. Therefore, it was challenging to increase the sample size in this retrospective study. Thus, a larger prospective multicenter study could be considered to confirm the result of the current study. 

We have revised the relevant sentences as shown below. 

Page 22, line 369 to line 370 (revised portion is in red font).

“Taken together, our findings need to be confirmed in a larger prospective multicentric study.”

4. Why H358 (EGFR WT) cells show a definite false positive signal utilizing the EGFR L858R antibody under Fig.S1A, in the lower panel? In such a scenario, how did authors claim their immuno-wall detection methodology to be free from any significant errors; that’s why limiting sample size is always deceptive!

Response: Thank you for your valuable comment. As mentioned, H358 cells showed weak fluorescence under Fig.S1A, indicating cross-reactivity of anti-EGFR (L858R) antibody. Hence, to estimate potential cross-reactivity, we performed the immunoassay using lysates from only the EGFR WT cell line (H358) (S2 Fig) and found that the fluorescence intensities remained below the threshold. In addition, as commented by reviewer #1, HCC827 cells showed weak fluorescence (Fig.S1B). We also performed the immunoassay using lysates from only HCC827 and found that the fluorescence intensities remained below the threshold, indicating the robustness of the assay even in samples with confirmed cross-reactivity by the anti-EGFR (L858R) antibody. Regarding the limited sample size in this study, as you have mentioned, selection bias might have affected the results. Thus, a larger prospective multicenter study could be considered to confirm the result of the current study.

Together with the addition of a new supplementary figure S2C, we have revised and included the following sentences as shown below. 

Page 13 line 225 to page 13 line 229 (revised portion is in red font).

“For L858R substitution antibody, western blot analysis revealed a weak band for L858R substitution in HCC827 cells, which also indicated cross-reactivity (S1 Fig). Therefore, immunoassay was performed using lysates from only HCC827 and observed that the fluorescence intensities remained below the threshold (S2C Fig).”

Page 19 line 322 to line 329 (revised portion is in red font).

“Moreover, the L858R substitution antibody might cross-react with WT EGFR or other EGFR mutations (S1 Fig). In previous studies where EGFR mutation was evaluated using IHC, false positive results were observed with the use of anti-EGFR (L858R) antibody [23,26]. To evaluate potential cross-reactivity, we performed the immunoassay using lysates from only HCC827, which were proven to be false positive by immunocytochemistry or western blotting (S1 Fig). We found that the fluorescence intensities remained below the threshold, indicating the robustness of the assay even in samples with confirmed cross-reactivity by the anti-EGFR (L858R) antibody.”

Page 30 line 504 to line 509 (revised portion is in red font).

“S2 Fig. Calibration curves of total protein concentration of lysates of EGFR WT cells (H358) and EGFR mutation cells (HCC827) versus the fluorescence intensity for each EGFR mutant protein. E746_A750 deletion for EGFR WT cell line (H358) (A), L858R substitution for EGFR WT cell line (H358) (B), and EGFR mutation cells (HCC827) (C). Dashed lines indicate three standard deviations above the average fluorescence determined in the EGFR WT cell line (H358) by each immuno-wall device.”

Page 22, line 369 to line 370 (revised portion is in red font).

“Taken together, our findings need to be confirmed in a larger prospective multicentric study.”

5. Why can’t authors patent their innovation before publishing it?

Response: Thank you for your valuable suggestion. We have already applied for a patent on our immunoassay. 

Other changes:

#1. Due to substantial contributions to the experiments required for this revision, Dr Hirofumi Shibata, Dr Kazuki Komeda, Dr Nozomi Kawabe, and Dr Kohei Matsuoka have been included as co-authors.

References

1. Kawahara A, Yamamoto C, Nakashima K, Azuma K, Hattori S, Kashihara M, et al. Molecular diagnosis of activating EGFR mutations in non-small cell lung cancer using mutation-specific antibodies for immunohistochemical analysis. Clinical cancer research : an official journal of the American Association for Cancer Research. 2010;16: 3163–70. doi:10.1158/1078-0432.CCR-09-3239

2. Seo AN, Park T-II, Jin Y, Sun P-LL, Kim H, Chang H, et al. Novel EGFR mutation-specific antibodies for lung adenocarcinoma: highly specific but not sensitive detection of an E746_A750 deletion in exon 19 and an L858R mutation in exon 21 by immunohistochemistry. Lung cancer (Amsterdam, Netherlands). 2014;83: 316–23. doi:10.1016/j.lungcan.2013.12.008

3. Eberhard DA, Giaccone G, Johnson BE, Group N-S-CLCW. Biomarkers of Response to Epidermal Growth Factor Receptor Inhibitors in Non–Small-Cell Lung Cancer Working Group: Standardization for Use in the Clinical Trial Setting. J Clin Oncol. 2008;26: 983–994. doi:10.1200/jco.2007.12.9858

4. Pirker R, Herth FJF, Kerr KM, Filipits M, Taron M, Gandara D, et al. Consensus for EGFR Mutation Testing in Non-small Cell Lung Cancer: Results from a European Workshop. J Thorac Oncol. 2010;5: 1706–1713. doi:10.1097/jto.0b013e3181f1c8de

5. Oliner K, Juan T, Suggs S, Wolf M, Sarosi I, Freeman DJ, et al. A comparability study of 5 commercial KRAS tests. Diagn Pathol. 2010;5: 23. doi:10.1186/1746-1596-5-23

6. Goto K, Satouchi M, Ishii G, Nishio K, Hagiwara K, Mitsudomi T, et al. An evaluation study of EGFR mutation tests utilized for non-small-cell lung cancer in the diagnostic setting. Annals of oncology : official journal of the European Society for Medical Oncology / ESMO. 2012;23: 2914–9. doi:10.1093/annonc/mds121

7. Xiong Y, Bai Y, Leong N, Laughlin TS, Rothberg PG, Xu H, et al. Immunohistochemical detection of mutations in the epidermal growth factor receptor gene in lung adenocarcinomas using mutation-specific antibodies. Diagnostic pathology. 2013;8: 27. doi:10.1186/1746-1596-8-27

8. Kato Y, Peled N, Wynes MW, Yoshida K, Pardo M, Mascaux C, et al. Novel epidermal growth factor receptor mutation-specific antibodies for non-small cell lung cancer: immunohistochemistry as a possible screening method for epidermal growth factor receptor mutations. Journal of thoracic oncology : official publication of the International Association for the Study of Lung Cancer. 2010;5: 1551–8. doi:10.1097/JTO.0b013e3181e9da60

9. Kitamura A, Hosoda W, Sasaki E, Mitsudomi T, Yatabe Y. Immunohistochemical Detection of EGFR Mutation Using Mutation-Specific Antibodies in Lung Cancer. Clinical Cancer Research. 2010;16: 3349–55. doi:10.1158/1078-0432.CCR-10-0129

10. Simonetti S, Molina M, Queralt C, Aguirre I de, Mayo C, Bertran-Alamillo J, et al. Detection of EGFR mutations with mutation-specific antibodies in stage IV non-small-cell lung cancer. Journal of Translational Medicine. 2010;8: 1–8. doi:10.1186/1479-5876-8-135

11. Brevet M, Arcila M, Ladanyi M. Assessment of EGFR mutation status in lung adenocarcinoma by immunohistochemistry using antibodies specific to the two major forms of mutant EGFR. The Journal of molecular diagnostics : JMD. 2010;12: 169–76. doi:10.2353/jmoldx.2010.090140

12. Wu S-GG, Chang Y-LL, Lin J-WW, Wu C-TT, Chen H-YY, Tsai M-FF, et al. Including total EGFR staining in scoring improves EGFR mutations detection by mutation-specific antibodies and EGFR TKIs response prediction. PloS one. 2011;6: e23303. doi:10.1371/journal.pone.0023303

13. Kozu Y, Tsuta K, Kohno T, Sekine I, Yoshida A, Watanabe S, et al. The usefulness of mutation-specific antibodies in detecting epidermal growth factor receptor mutations and in predicting response to tyrosine kinase inhibitor therapy in lung adenocarcinoma. Lung cancer (Amsterdam, Netherlands). 2011;73: 45–50. doi:10.1016/j.lungcan.2010.11.003

14. Hofman P, Ilie M, Hofman V, Roux S, Valent A, Bernheim A, et al. Immunohistochemistry to identify EGFR mutations or ALK rearrangements in patients with lung adenocarcinoma. Annals of Oncology. 2012;23: 1738–1743. doi:10.1093/annonc/mdr535

15. Ikami M, Kawakami A, Kakuta M, Okamoto Y, Kaji N, Tokeshi M, et al. Immuno-pillar chip: a new platform for rapid and easy-to-use immunoassay. Lab on a chip. 2010;10: 3335–40. doi:10.1039/c0lc00241k

---

## [Editor Report · Decision Letter 1]

15 Oct 2020

Development of an immuno-wall device for the rapid and sensitive detection of EGFR mutations in tumor tissues resected from lung cancer patients

PONE-D-20-01970R1

Dear Dr. Hase,

We’re pleased to inform you that your manuscript has been judged scientifically suitable for publication and will be formally accepted for publication once it meets all outstanding technical requirements.

Kind regards,

Srikumar Chellappan

Academic Editor

PLOS ONE
---

## [Editor Report · Acceptance letter]

6 Nov 2020

PONE-D-20-01970R1 

Development of an immuno-wall device for the rapid and sensitive detection of *EGFR* mutations in tumor tissues resected from lung cancer patients 

Dear Dr. Hase:

I'm pleased to inform you that your manuscript has been deemed suitable for publication in PLOS ONE. Congratulations! Your manuscript is now with our production department. 

Kind regards, 

on behalf of

Dr. Srikumar Chellappan 

Academic Editor

PLOS ONE